# Study on the Corrosion Resistance of Laser Clad Al$_{0.7}$FeCoCrNiCu$_x$ High-Entropy Alloy Coating in Marine Environment

**Xuehong Wu** [1,2,3,*] **and Yanjun Lu** [4]

1  Scientific Research Office, Gansu Vocational &Technical of Nonferrous Metallurgy, Jinchang 737100, China
2  School of Mechanical & Electrical Engineering, Lanzhou University of Technology, Lanzhou 737050, China
3  Engineering Research Center for New Equipment of Non-Ferrous Metallurgy, Ministry of Education, Lanzhou 737050, China
4  School of Mechanical and Precision Instrument Engineering, Xi'an University of Technology, Xi'an 710048, China
*  Correspondence: wxh3215550@163.com; Tel.: +86-15097028894

**Abstract:** In the marine atmosphere, the corrosion rate of ship components is 4–5 times higher than that of the inland atmosphere. To solve the serious corrosion problem arising from long-term service in the marine environment of naval aircraft and ships, etc., this paper takes Al$_{0.7}$FeCoCrNiCu$_x$ system high-entropy alloy coating prepared by laser melting technology with 5083 aluminum alloys as the base material and analyzes the aging and failure mode of equipment coating under a marine atmospheric environment. XRD and SEM were utilized to study the microscopic morphological structure of the coatings. The laws of influence of Cu elements on the electrochemical corrosion behavior of the Al$_{0.7}$FeCoCrNiCu$_x$ system high-entropy alloy in 3.5 wt.% NaCl neutral solution was investigated by using dynamic potential polarization and electrochemical impedance spectroscopy, and neutral salt spray acceleration tests and outdoor atmospheric exposure tests were carried out. The results show that the Al$_{0.7}$FeCoCrNiCu$_x$ ($x = 0$) high-entropy alloy coating has a single BCC phase structure and the Al$_{0.7}$FeCoCrNiCu$_x$ ($x = 0.30, 0.60, 0.80, 1.00$) high-entropy alloy coating consists of both BCC and FCC phases with a typical dendrite morphology. With the increase in Cu content, the self-corrosion potential of Al$_{0.7}$FeCoCrNiCu$_x$ gradually increases and the current density gradually decreases, which with the results of the electrochemical impedance spectrum analysis, indicating that the corrosion resistance of Al$_{0.7}$FeCoCrNiCu$_{1.00}$ is optimal. The results of the neutral salt spray acceleration test and the outdoor atmospheric exposure test were integrated to conduct a comprehensive evaluation of the corrosion resistance of the coating. The corrosion resistance of Al$_{0.7}$FeCoCrNiCu$_x$ coating increases with the increase in Cu content, and the impressive strength and plastic deformation are best when $x = 0.80$. Neutral salt spray accelerated the test with no corrosion at 5040 h, and even if the coating is broken, it can last up to 4320 h. In the outdoor atmospheric exposure test, which was conducted 12 months after the coating surface test, no corrosion occurred.

**Keywords:** high-entropy alloy coating; marine atmospheric environment; corrosion resistance; neutral salt spray acceleration test; outdoor exposure experiment





## 1. Introduction

With the emergence of new marine industries such as marine transportation, deep-sea mining, port terminals, oil and gas development, and marine biotechnology, etc., the scale of human exploitation of the ocean is expanding and gradually moving from traditional to modern. In this process, the progress of anti-corrosion coatings and their technologies for ships and other maritime equipment has become the focus of research. The degree of corrosion of the marine environment from ships and other marine equipment directly determines the environmental adaptability of the ship, the reliability of its use, and

the life of its hull, and even affects its combat effectiveness and revivability at sea. The average salinity of seawater is 3.5 wt.%, plus the presence of some other impurities and metal ions, transforming seawater into a strong electrolyte. Materials such as aluminum, steel, and other metals are prone to galvanic corrosion, crevice corrosion, stress corrosion, intergranular corrosion, etc., resulting in poor institutional strength and maneuverability of the ship and leading to a reduction in propeller propulsion efficiency, noise increases, instrumentation failure, interference with sonar, etc. This not only shortens the service life of the ship and reduces the in-service rate, but also increases the ship's sailing resistance, thus, significantly increasing fuel consumption [1]. When subject to different seasons or regions, submerged protective coatings may be exposed to harsh atmospheric conditions such as high-temperature exposure, low-temperature freezing, and strong winds, accelerating the damage incurred; therefore, the use of surface engineering technology the strengthen the surface of key ship components is a proven method [2]. High-energy beam laser melting technology [3] has the advantages of a fast cooling rate, small heat-affected zone, and small substrate deformation, and the prepared coating has a low dilution rate, fine and uniform dense organization, and forms a good metallurgical bond with the substrate; thus, it is widely used for the surface strengthening of large machinery and key components in the fields of shipping tools, metallurgy, and mining. $Al_{0.7}$FeCoCrNi high-entropy alloys [4], as common multi-principal systems, are highly promising structural materials because of their excellent properties such as high strength and hardness, superior high-temperature oxidation resistance, and outstanding corrosion resistance [5]. Ti, Cu, and Zr, as large atomic radius elements, are typically used as additive elements to enhance the comprehensive performance of high-entropy alloys. The addition of a small amount of Cu to $Al_{0.7}$FeCoCrNi results in a smaller grain size, a higher integral number of precipitated objects, and significantly increases the plasticity of the material [6].

D. V. Mashtalyar et al. [7] used a combination of the plasma electrolytic oxidation (PEO) method and fluoropolymer spraying method to prepare polymer coatings of alloys on Mg-Mn-Ce alloys. The corrosion current density was three orders of magnitude lower than that of the basic PEO layer and six orders of magnitude lower than that of the magnesium alloy, and the corrosion resistance was significantly improved. In the potentiodynamic mode, the protective coating is formed on Mg-Mn-CE alloy by the PEO method. The coating with considerable porosity can reduce its polarization resistance, increase the corrosion current density, and reduce the corrosion resistance [8]. Liu et al. [9] prepared sol-gel coating, epoxy resin sealing coating, and post-treated AM50 magnesium alloy PEO coating. It is concluded that the sol-gel coating and the epoxy resin composite coating can effectively close the openings of the PEO coating and greatly improve its corrosion resistance. Moreover, the longer the post-treatment time, the better and more stable the corrosion resistance of the epoxy resin composite sealing coating than that of the sol-gel coating. In this paper, the laser cladding technology is used to form a metallurgical bond between the corrosion-resistant and high-temperature oxidation-resistant high-entropy alloy coating and the surface of the ship parts. The corrosion resistance of the $Al_{0.7}$FeCoCrNiCu$_x$-based high-entropy alloy coatings was studied from various aspects, such as microstructure, corrosion resistance reasons, and electrochemical properties, and verified by a neutral salt spray accelerated test and an outdoor atmospheric exposure test.

To study the corrosion behavior of Ni element on Zr-based amorphous alloy in a Cl-environment, Gan et al. [10] selected $Zr_{55}Cu_{30}Ni_5Al_{10}$ and $Zr_{55}Cu_{35}Al_{10}$ bulk amorphous alloys and investigated the effect of Ni element on Zr-Cu-Al system amorphous alloy in 3.5 wt.% NaCl neutral solution, using dynamic potential polarization and electrochemical impedance spectroscopy. The influence law of Ni element on the electrochemical corrosion behavior of the Zr-Cu-Al system amorphous alloy in NaCl solution was comparatively examined. It is noted that $Zr_{55}Cu_{30}Ni_5Al_{10}$ containing Ni element has better corrosion resistance than $Zr_{55}Cu_{35}Al_{10}$ amorphous alloy in NaCl solution. Zr-Cu-Al amorphous alloys undergo pitting corrosion in NaCl solution, and the circular corrosion pits are full of foam-like pores. Through comparison of the element distribution before and after

corrosion, it is found that the alloy elements of Zr-Cu-Al amorphous alloy are selectively dissolved under the action of $Cl^-$. The amorphous alloy containing Ni element constitutes a dense passivation film, which inhibits the selective dissolution of metal elements, thereby improving corrosion resistance.

The team of Yang [11] systematically described the mechanisms related to the influence of Cu elements on the microstructure, interfacial behavior, phase evolution, and electrochemical corrosion performance of Ni-based alloy coatings with oriented structures. The research shows that the content of Cu element has a significant effect on the grain growth behavior, grain boundary morphology, and composition of the coating, which leads to the transformation of the coating's electrochemical corrosion mechanism in 10% sulfuric acid and significantly affects its corrosion resistance. The trace element of Cu can weaken the anodic reaction behavior of the coating surface and significantly improve the coating's corrosion resistance, but excessive Cu element will lead to Cu enrichment at the grain boundary, forming a regional corrosion galvanic cell and promoting the pitting corrosion behavior of the coating.

Zhang et al. [12] investigated the frictional wear behavior of AlNiZr amorphous nanocrystalline composite coatings prepared by high-speed arc spraying in a NaCl solution with a mass fraction of 3.5 wt.%. The matrix material selected by the researchers is commercial 45 steel (medium carbon quenched and tempered steel), and the raw material used for high-speed arc spraying is AlNiZr powder core wire (w(Ni) = 20%~30%, w(Zr) = 3%~10%; the rest is Al). The study found that the structure of AlNiZr coating is relatively uniform and dense; the phase structure is composed of amorphous, nanocrystalline, and crystalline phases; the amorphous fraction of the coating is about 64.93%; the average microhardness value is 363 $HV_{0.1}$; and the average bonding strength with the 45 steel substrate is about 30.8 MPa. Under dry friction conditions, the average friction coefficient is about 0.125, the wear volume is about 0.134 $mm^3$, and the wear scar width is about 882.4 μm. The wear failure mechanism is mainly oxidative wear and brittle peeling wear, accompanied by slight abrasive wear. Under the condition of a corrosive medium, due to the lubrication and friction reduction effect of the corrosive medium, the average friction coefficient, wear volume, and wear scar width of the coating is significantly reduced. The average friction coefficient is about 0.058, the wear volume is about 0.02216 $mm^3$, and the width of the wear scar is about 314 μm. The failure mechanism of corrosion wear is mainly in the form of delamination wear, and wear plays a leading role, followed by corrosion. Regarding pure aluminum coating, AlNiZr coating exhibits excellent corrosion and wear resistance.

Liu et al. [13] studied the rapid solidification and liquid phase separation of supercooled CoCrCuFe$_x$Ni high-entropy alloys and concluded that the mixing enthalpies of Cu-Fe, Cu-Co, Cu-Cr, and Cu-Ni are +13, +6, +12, and +4 KJ/mol, which are all positive mixing enthalpies. Therefore, from the point of view of thermodynamics, it is difficult for Cu to create a strong bonding force with Fe, Co, Cr, and Ni to form a uniform solid solution. Cu-containing high-entropy alloys tend to form Cu-rich and Cu-poor phases, reducing the electrochemical corrosion resistance of the alloy.

Previous studies on the effects of coating preparation methods and elements on the mechanical properties (such as hardness, plasticity, wear resistance, wear reduction, etc.) of high-entropy alloys are more numerous, and the coatings prepared with different process parameters had defects such as a sparse microstructure and large grain size. However, among the areas rarely reported are the study of the corrosion resistance mechanism of a single element on high-entropy alloy coatings, and the design and implementation of outdoor experiments to verify the results of corrosion resistance experiments of high-entropy alloy for a long period. Moreover, the data of only a few tests are not enough to support the study of ships' corrosion resistance in the freshwater environment. In the actual service process of ships, corrosion resistance is an important index affecting their service life, so the preparation of coatings with excellent mechanical properties and corrosion resistance is of great importance to national defense construction. In this paper, the Al$_{0.7}$FeCoCrNiCu$_x$ ($x$ is the ratio of a substance, $x$ = 0.00, 0.30, 0.60, 0.80, 1.00)

high-entropy alloy coating is prepared on the surface of Al alloy using laser melting technology, and metallurgical bonding is formed with the surface of naval components to study the corrosion resistance of the $Al_{0.7}FeCoCrNiCu_x$ high-entropy alloy in terms of its metallography, coating microstructure, corrosion resistance mechanism, electrochemistry, and other aspects. The corrosion resistance of the $Al_{0.7}FeCoCrNiCu_x$ high-entropy alloy was studied in terms of metallography, coating microstructure, corrosion resistance mechanism, electrochemical corrosion resistance, etc. The neutral salt spray accelerated test and outdoor atmospheric exposure test were also designed to verify the corrosion resistance of the coating under a real service environment to provide a theoretical and experimental basis for improving the corrosion resistance of naval components.

## 2. Aging and Failure Modes of Equipment Coatings in Marine Environments

Protective coating is an important tool to ensure the corrosion resistance of equipment. The equipment serving in the marine environment is corroded to varying degrees after the aging and failure of the protective coating [14]. Therefore, through the investigation of equipment corrosion, the aging failure mode of the protective coating and the weak link of coating protection was found. According to the investigation, the weak links of protection and the aging failure modes of coatings for three types of marine environmental equipment were analyzed, as showed in Table 1.

**Table 1.** Aging failure modes of equipment coatings serviced in marine environments.

| Equipment Type | Weak Link of Protection | Aging Failure Mode of Coating |
|---|---|---|
| Outdoor communication equipment | Fastener connection | Cracking, peeling of coating, and corrosion of base metal |
| | Support gap | Coating peeling, substrate metal corrosion |
| | Protective cover | Surface coating pulverization |
| | Dissimilar material joint | Coating peeling, substrate metal corrosion |
| | Edges and welding parts of structural members | Coating stress concentration, spalling, and substrate metal corrosion |
| | Printed circuit board | Coating mildew, solder joint and line corrosion |
| | Component surface coating | Coating cracking and falling off |
| Vehicle | Engine water tank | Coating stress concentration, spalling, and substrate metal corrosion |
| | Component edge | Coating stress concentration, spalling, and substrate metal corrosion |
| | Fastener connection | Cracking and peeling of coating and corrosion of base metal |

## 3. Experimental Materials and Methods

### 3.1. Experimental Equipment and Materials

The equipment and parameters involved in this experiment are shown in Table 2.

**Table 2.** The experimental equipment.

| Equipment Name | Model | Brand | Manufacturer | Producing Country |
|---|---|---|---|---|
| Planar $CO_2$ laser | DC050 | ROFIN | ROFIN | Germany |
| Vertical ball mill | AX-100 | Haibo | Wuxi Haibo powder equipment Co., Ltd. | China |
| Double-barrel powder feeder | DPSF-2 | Everest | Jiangsu Everest Laser Technology Co., Ltd. | China |

**Table 2.** *Cont.*

| Equipment Name | Model | Brand | Manufacturer | Producing Country |
|---|---|---|---|---|
| X-ray diffractometer | Empyrean | Sharp image | PANalytical | Netherlands |
| Electrochemical workstation | Zennium X | Zennium | Zana company | Germany |
| Salt spray corrosion test chamber | Q-FOGCCT | Q-Lab | Q-Lab | USA |

In this experiment, 5083 (melted and then cooled—the material was artificially aged without any plastic deformation) aluminum alloy was the base material, the size was 50 mm × 30 mm × 10 mm, and its chemical composition is shown in Table 3. There was an aluminum-clad layer on the surface of the original material, and this was removed by etching with NaOH solution.

**Table 3.** Chemical composition of 5083 aluminum alloy (wt.%).

| Element | Mg | Si | Cu | Zn | Mo | Ti | Cr | Al |
|---|---|---|---|---|---|---|---|---|
| Content | 3.8 | 0.5 | 0.2 | 0.3 | 1.3 | 0.3 | 0.06 | 93.54 |

*3.2. Experimental Process*

3.2.1. Preparation of High-Entropy Alloy

The oxide film of the substrate should be removed with sandpaper before laser deposition. After cleaning, it should be placed into a vacuum-drying oven for testing. To ensure that the powder had good fluidity in the powder feeder, spherical Al, Cr, Fe, Co, Ni, and Cu metal powders with a purity greater than 99.9% were selected [15].

Five $Al_{0.7}FeCoCrNiCu_x$ ($x$ = 0, 0.30, 0.60, 0.80, 1.0) coatings with different copper contents were designed. These alloys are abbreviated as $Cu_0$, $Cu_{0.30}$, $Cu_{0\ 60}$, $Cu_{0.80}$, and $Cu_{1.00}$. The specific powder ratio and alloy abbreviations are shown in Table 4.

**Table 4.** Specific powder ratios and alloy abbreviations (at%).

| Abbreviation | Al | Cr | Fe | Co | Ni | Cu |
|---|---|---|---|---|---|---|
| $Cu_0$ | 15.59 | 20.56 | 19.33 | 19.66 | 19.84 | 0 |
| $Cu_{0.30}$ | 14.78 | 18.95 | 18.76 | 18.75 | 18.79 | 4.73 |
| $Cu_{0.60}$ | 14.93 | 18.77 | 17.35 | 17.88 | 17.68 | 8.75 |
| $Cu_{0.80}$ | 13.95 | 17.95 | 17.44 | 17.59 | 17.05 | 12.56 |
| $Cu_{1.00}$ | 12.88 | 16.70 | 16.37 | 16.34 | 16.77 | 16.32 |

The powder was stirred with a vertical ball mill under an argon atmosphere for 2 h and then kept in a vacuum drying oven at 80 °C for 2 h. A double-cylinder powder feeder was selected, and the carrier gas was argon. Based on a large number of previous experiments, the pulsed laser deposition mode was selected. The specific parameters were set as follows: the spot diameter was 1.2 mm, the laser power was 1850 W, the powder feeding speed was 6.2 g/min, the scanning speed was 115 mm/min, the duty cycle was 75%, the pulse frequency was 65 Hz, the overlap rate was 35%, and the carrier gas flow rate was 6 L/min [16].

Phase detection was performed using an X-ray diffractometer. The scanning speed was 4°/min. The scanning step was 0.05°, and the scanning interval was 20° to 90°. The metallographic samples were prepared along the direction perpendicular to the scanning speed and corroded with a mixture of hydrochloric acid and nitric acid at a concentration of 3:1, and the microstructure of the material was observed by scanning electron microscope.

An electrochemical workstation was used with 3.5% NaCl as the electrolyte, the working electrode as the sample, the counter electrode as the platinum electrode, the reference electrode as the saturated calomel electrode, the relative scanning potential range at −3 V to 3 V, and the scanning rate at 5 mV/s. The frequency range of the electrochemical impedance spectroscopy was 100 mHz to 100 kHz, the open circuit potential was 10 mV, and the morphology after corrosion was photographed by scanning electron microscope.

### 3.2.2. Neutral Salt Spray Acceleration Test and Outdoor Atmospheric Exposure Test

The neutral salt spray accelerated test was carried out in the salt spray corrosion test chamber according to the regulations of ASTM B117. The neutral solution was prepared with industrial sodium chloride with a purity of ≥95.5% as the corrosion accelerated solution, with a pH of 6.5 to 7.2, and pressurized into a fine mist to evenly distribute it on the sample surface [17].

The test conditions were as follows: salt solution, NaCl solution with a concentration of 3.5 wt.% by mass, and glacial acetic acid to adjust the pH to 3.1 to 3.3 (the content of glacial acetic acid was 0.1% to 0.3%). The temperature in the test chamber was kept at (35 + 1) °C, and the specimen was placed at an angle of 45° to the vertical direction; the spray volume was controlled within the range of 1.0–2.0 mL/(80 cm$^2$·h). After the accelerated corrosion test, the specimens were analyzed and evaluated. The corrosion degree was evaluated by the method of measuring weight loss, and the mechanical properties were measured and compared. At the same time, the 5083 aluminum alloy substrate was added for comparison with the coating, and the corrosion of the salt spray was regularly observed. Under the provisions of ASTM G50, outdoor exposure experiments were carried out in Beijing outdoor exposure field and Qingdao Tuandao outdoor exposure field, respectively, and the corrosion resistance of the laser cladding $Al_{0.7}FeCoCrNiCu_{1.00}$ coating was comprehensively tested.

In order to comprehensively investigate the corrosion resistance of the coating, the $Al_{0.7}FeCoCrNiCu_{1.00}$ high-entropy alloy coating and 5083 aluminum alloy substrates were selected for comparative experiments. Figure 1 shows the appearance of 5083 aluminum alloy and $Al_{0.7}FeCoCrNiCu_{1.00}$ high-entropy alloy coating before the experiment.

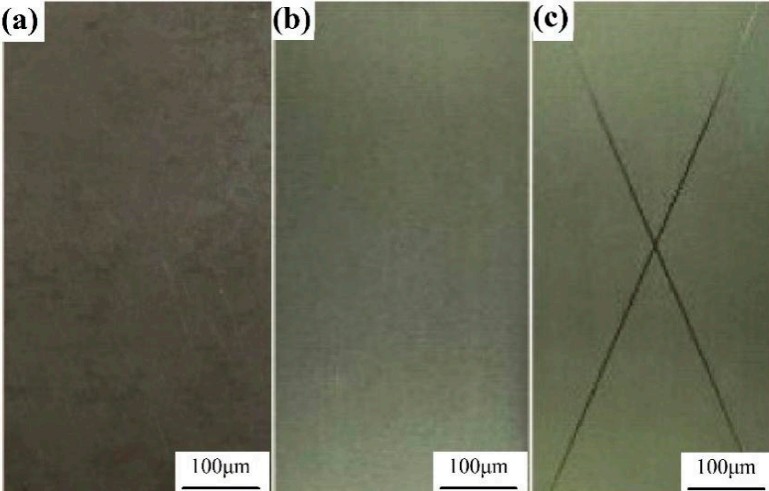

**Figure 1.** 5083 aluminum alloy and $Al_{0.7}FeCoCrNiCu_{1.00}$ high-entropy alloy coating. (**a**) 5083 aluminum alloy; (**b**) $Al_{0.7}FeCoCrNiCu_{1.00}$ high-entropy alloy coating; (**c**) cross marking on $Al_{0.7}FeCoCrNiCu_{1.00}$ high-entropy alloy coating.

## 4. Results and Analysis

### 4.1. XRD Diffraction Results and Analysis of Coating

The XRD pattern of the $Al_{0.7}FeCoCrNiCu_x$ coating is shown in Figure 2. $Al_{0.7}FeCoCrNiCu_x$ is composed of the BCC1 and BCC2 phases and the FCC phase. Among them, the BCC1 structure is composed of CrNi, AlNi, $CrNi_2$, etc.; the BCC2 structure is composed of $Fe_3Cr_2$, $Fe_2Cr$, $Co_3Ni_2Cu$, and other structures; and the FCC phase is mainly composed of $FeNi_3$ and $FeNi_2$. Figure 2 shows that when $x = 0$, the coating only contains the BCC phase, and there is no significant FCC phase diffraction peak. This agrees with the results of AGAR-WAL [18] and is due to the high entropy effect, hysteresis diffusion, and rapid cooling of the laser melting, which is characteristic of high-entropy alloys, forming a supersaturated solid solution and inhibiting the formation of intermetallic compounds during solidification. When $x = 0.3$, the diffraction peak of the intermetallic compound phase in the coating disappears. The addition of the Cu element increases the mixing entropy of the multi-principal alloy system, which is conducive to the formation of simple solid solution structures of BCC and FCC, so the FCC phase gradually increases. When $x = 0.6$, it can be judged from the intensity of diffraction peaks that the content of the FCC phase increases significantly, while the content of the BCC phase decreases. It is shown that the addition of the Cu element promotes the formation of the face-centered cubic structure while suppressing the formation of the body-centered cubic structure. When $x = 0.8$, the intensity of the BCC diffraction peak is significantly weakened, the FCC phase gradually increases and reaches the maximum amount, and the BCC phase gradually decreases. When $x = 1.00$, the FCC phase gradually decreases, and the BCC phase gradually increases. Therefore, with the increase in Cu content, the relative content of FCC in the alloy coating first increases and then decreases, and at $x = 0.8$, the FCC phase has the most–even exceeding the content of the BCC phase—and dominates.

Judging from the intensity of the diffraction peaks, the BCC phase and the FCC phase are mainly concentrated around the diffraction angles of 44° and 45°. With the increase in Cu content, the solid solution degree of Cu atoms in the alloy gradually increases, and there is a certain difference in the size of each atom in the alloy component, which leads to the increase in lattice distortion and the shift of diffraction angle to a lower angle. The intensity of the FCC diffraction peak gradually decreases due to the diffuse reflection effect as the diffraction angle increases and the intensity of the diffraction peak decreases. [19,20].

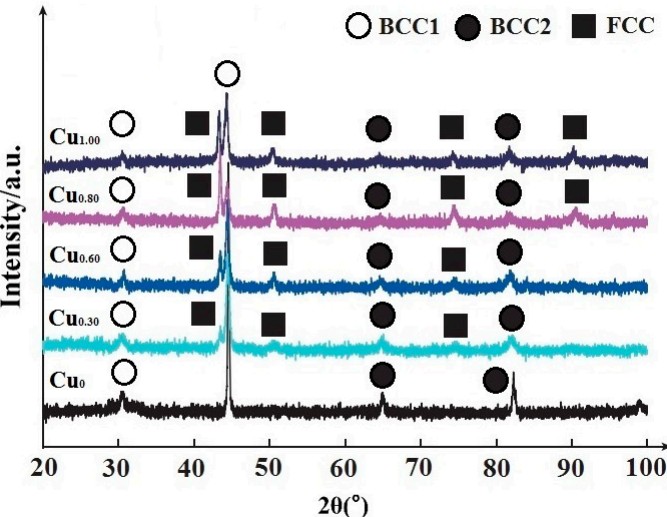

**Figure 2.** XRD pattern of high-entropy alloy.

The figure shows that the phase structure of laser cladding $Al_{0.7}FeCoCrNiCu_x$ high-entropy alloy coating comprises only body-centered cubic (BCC) phase and face-centered cubic (FCC) phase. This is because the atomic size difference between Fe, Co, Cr, and Ni is small, and the atomic binding energy between each element is also small, so the probability

of the element occupying each position of the crystal lattice to form a simple solid solution structure is the same. Moreover, the coating is composed of multiple principal elements, and the formation of a simple solid solution structure will greatly increase the mixing entropy of the system so that the free energy changes greatly when the alloy system is formed. These factors accord the high-entropy alloy the tendency to be stable. That is, multiple principal elements produce a high mixing entropy effect [21], causing the high-entropy alloy system to form a crystal structure and improving the stability of the system. The addition of a large atomic radius Cu is beneficial to the precipitation of BCC term in the alloy [22]. At the same time, the structure of the BCC phase is looser due to the large difference in atomic size and severe lattice distortion, thus, regulating the strain on the lattice and reducing the free energy of the system.

According to the Gibbs phase law, the number of principal components added in each set of experiments in this study is greater than or equal to 5, and the number of alloy phases is much smaller than the predicted value of the number of phases [23]. This is mainly because the mixing entropy of the alloy increases after mixing elements in close to equimolar molar ratio, which inhibits the formation of intermetallic compounds with complex phase structures in high-temperature reactions and makes it easier to form simple phase structures with lower free energy. The $Cu_0$ alloy exists only in the BCC1 phase. The diffraction peak near 31° proves the existence of an ordered BCC1 phase (AlNi), and the BCC2 phase is a multi-element (Fe-Cr) solution. When $x = 0.25$, the FCC phase diffraction peak appears in $Cu_{0.30}$ alloy. Near 51°, 75°, and 83°, the diffraction peaks of the FCC phase gradually increase with the increase in $x$, indicating that Cu can promote the formation of the FCC phase. In addition, it is found that the intensity of diffraction peaks at large angles is very weak in the XRD pattern. This is because, under the multi-element high-entropy alloy, the lattice distortion of different atomic sizes is larger, and the diffusion effect is gradually enhanced with the increasing angle, so the intensity of the diffraction peak is relatively weak [24].

### 4.2. Coating Microstructure

Figure 3 shows the microstructures of the $Al_{0.7}FeCoCrNiCu_x$ high-entropy alloy coatings with different Cu contents, with GM and GB representing the grain and intergranular regions.

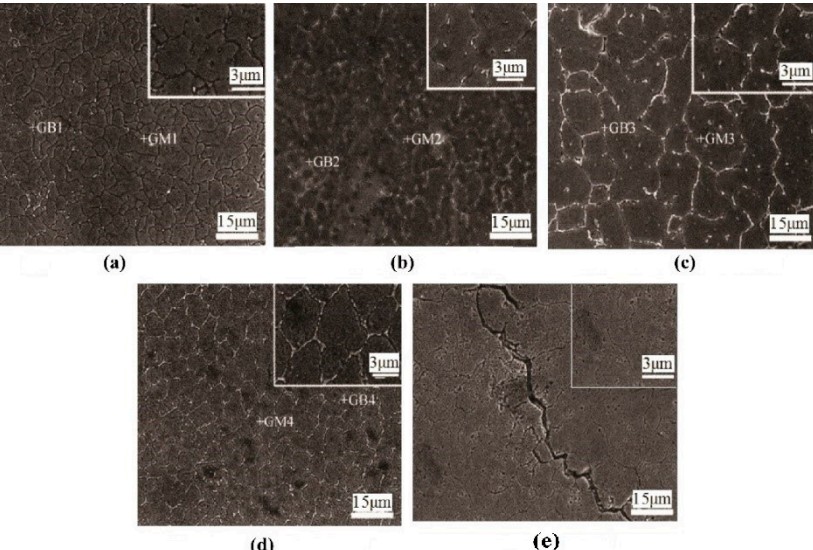

**Figure 3.** Microstructure of $Al_{0.7}FeCoCrNiCu_x$ high-entropy alloy coatings with different Cu con-tents. (**a**). $Al_{0.7}FeCoCrNiCu_{0.30}$; (**b**). $Al_{0.7}FeCoCrNiCu_{0.60}$; (**c**). $Al_{0.7}FeCoCrNiCu_{0.80}$; (**d**). $Al_{0.7}FeCoCrNiCu_{1.0}$; (**e**). $Al_{0.7}FeCoCrNi$.

Figure 3 shows the microstructures of the $Al_{0.7}FeCoCrNiCu_x$ high-entropy alloy coatings with different Cu contents. It can be seen that the $Al_{0.7}FeCoCrNi$ alloy suffers from microporous aggregation-type ductile fracture when no Cu element is added. $Al_{0.7}FeCoCrNiCu_{0.30}$ shows a large number of deconstruction scoring and secondary cracks; the white parts are dendritic and punctate organization, and the dendritic is coarser. Spherical grains are precipitated at the dendrite boundary with a size of a few ten nanometers. With the increase in Cu content, the grains gradually change from dendritic to dendritic and punctate, showing a typical dendritic structure. However, with the continuous increase in Cu content, the area outside the dendritic and punctate structures becomes larger. Due to the diffusion and redistribution of multiple elements during the solidification process of the alloy, the nucleation and growth of precipitates are hindered, and the formation of nanophases is favorable, which is a prominent feature of high-entropy alloys [25].

Figure 3a shows the microstructure of the $Cu_{0.30}$ alloy; the particles are petal-shaped and the surface has tiny corrosion holes. The distribution of copper elements in the crystal and among the grains is uniform. Figure 3b shows the microstructure of the $Cu_{0.60}$ alloy, a light-colored structure with dispersed distribution appears at the grain boundary, and copper segregation appears in the intergranular region. The distribution of other elements is similar to that of $Cu_{0.30}$ alloy. As shown in Figure 3c, when the copper atomic ratio is increased to 0.80, the cladding layer structure is composed of dendrite and a network eutectic structure. Since these two structures are related to each other, in the process of extending the eutectic grains from the center to the periphery, the flaky corrosion-resistant phase is always thin, and the flaky structure gradually becomes coarser when it is adjacent to the grain boundary. This may be due to the large latent heat of crystallization of the composition, which slows down the later solidification, and the grain structure becomes coarse. The adjacent eutectic grains grow and join at the same time, forming a network structure around the grain boundary [26]. As shown in Figure 3d, the morphology of the $Cu_{1.00}$ alloy is similar to that of the $Cu_{0.80}$ alloy, and copper is still segregated at the grain boundaries [27]. The grains and grain boundaries have been fused to form the atomic radius of the elements close to each other, and the interior of the alloy is more inclined to form a substitutional solid solution. When the solid solution in the alloy reaches saturation, the second-phase particles will segregate at the grain boundary, which will have a dispersion-strengthening effect on the matrix. As shown in Figure 3e, cracks have appeared in the $Cu_0$ alloy coating. The cracks are caused by the BCC phase structure of the $Cu_0$ alloy. This is because, although the hardness of the BCC phase is higher, it increases the brittleness of the alloy and is easy to fracture when the alloy is solidified.

Table 5 shows the energy spectrum analysis of the regions GM1–GM4 and GB1–GB4 in Figure 3. According to the Gibbs free energy theory, the segregation phenomenon of Cu is reasonably explained. The large differences in mixing enthalpy and mixing entropy between Cr, Fe, and Cu make it difficult for two elements, Cr and Fe, to coexist with Cu, while the equiaxed crystal structure of the Cu element itself determines that the ideal fusion of the other five elements cannot be achieved either. Therefore, the two factors of Cu equiaxed crystal structure and molar mixing enthalpy lead to serious segregation of Cu between grain boundaries. When the content of Cu element is low, the AlNi phase formed in the solid solution constitutes dendrites; with the continuous increase in Cu element content, the elements with a strong binding force to Cu remain inside the grains, and the elements with a weak binding ability are precipitated into grain boundaries and between dendrites.

**Table 5.** EDS analysis of zone GB1-GM4, GB1-GB4 in Figure 3 at %.

| Location | Al | Fe | Co | Cr | Ni | Cu |
|---|---|---|---|---|---|---|
| GB1 | 17.31 | 21.25 | 17.40 | 20.15 | 18.32 | 0.18 |
| GM1 | 12.48 | 24.36 | 13.54 | 28.63 | 11.61 | 1.36 |
| GB2 | 14.95 | 18.59 | 19.21 | 18.35 | 22.87 | 3.54 |
| GM2 | 13.87 | 20.14 | 17.05 | 20.31 | 18.74 | 6.15 |
| GB3 | 11.37 | 18.65 | 16.68 | 21.32 | 18.68 | 9.32 |
| GM3 | 18.45 | 14.48 | 14.35 | 20.54 | 16.25 | 12.78 |
| GB4 | 7.75 | 13.50 | 21.20 | 21.46 | 19.36 | 15.85 |
| GM4 | 11.37 | 17.95 | 22.08 | 18.21 | 17.15 | 21.72 |

According to the energy spectrum analysis in Table 5, it can be concluded that the solidification process of the alloy is as follows. When the alloy begins to solidify from the liquid phase, the high-melting primary phase rich in ($\alpha$-Fe, Cr) first precipitates from the liquid phase, and then the Al, Ni, Co phases adhere to the primary phase of ($\alpha$-Fe, Cr) for nucleation. Then, a two-phase alternating structure is gradually formed, and a eutectic group grows from a crystal nucleus; at the same time, the Cu element is arranged into the intergranular. Due to the obvious chilling effect of the air, the growth rate of the eutectic group is fast, and a layer of fine eutectic cells is formed on the outside. Then, the eutectic cell inside the liquid phase grows gradually, and the core finally nucleates. Due to the long growth time and low nucleation rate, the eutectic cell in the core is the coarsest. As the effect of mixing entropy on the stability of solid solution decreases as the temperature decreases, alloys often undergo phase transformations during solidification, such as destabilizing decomposition, ordering, or precipitation. However, the long-range diffusion in the phase separation process of solid multi-principal alloys is slow because there is no main matrix element, and the substitutional diffusion of elements in the alloy is difficult. Coupled with the interaction of diffusing particles during distribution, the nucleation rate and growth rate of crystals are greatly reduced, so that multi-component alloys will form nanostructures.

*4.3. Corrosion Resistance of Coating*

The electrochemical corrosion morphology of the $Al_{0.7}FeCoCrNiCu_x$ high-entropy alloy coating in 3.5% NaCl solution is shown in Figure 4.

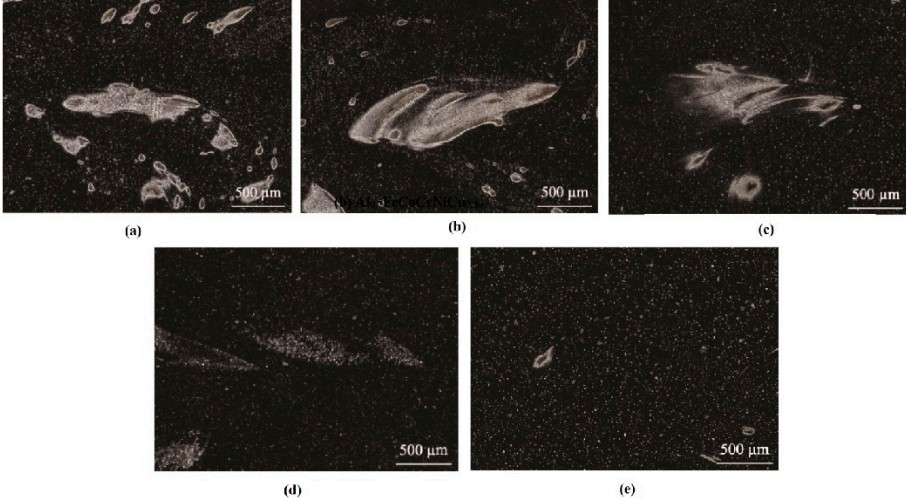

**Figure 4.** SEM images of corrosion morphology of $Al_{0.7}FeCoCrNiCu_x$ high-entropy alloy coatings with different Cu contents. (**a**). $Al_{0.7}FeCoCrNi$; (**b**). $Al_{0.7}FeCoCrNiCu_{0.30}$; (**c**) $Al_{0.7}FeCoCrNiCu_{0.60}$; (**d**). $Al_{0.7}FeCoCrNiCu_{0.80}$; (**e**). $Al_{0.7}FeCoCrNiCu_{1.00}$.

Figure 4 is the SEM photo of the corrosion morphology of the Al$_{0.7}$FeCoCrNiCu$_x$ high-entropy alloy coating. Figure 4a shows that when no Cu element is added, corrosion pits with different diameters and irregular shapes appear after the polarization reaction. Most of the corrosion pits are uneven and have densely distributed foam-like holes with a small structure and different sizes. Foam-like holes are also distributed around the corrosion pit; the corrosion area is relatively large, and the outer edge expands outward, showing a "river flow" shape. Microcracks and flaky structures are distributed in part of the corrosion pit (in the red frame). As the content of Cu element gradually increases, as can be seen in Figure 4b, the area of the corrosion pit gradually decreases, and the internal cracks are gradually replaced by foam-like pores. As can be seen in Figure 4c,d, with the further increase in Cu element content, the degree of reduction in the corrosion pit area is obvious. In Figure 4e, the corrosion pits with a relatively large area are rarely seen and the size of the corrosion pits is smaller, the pitting corrosion is discontinuously distributed, and the surrounding of the corrosion pits is relatively smooth.

Figure 5 shows a schematic diagram of the electrochemical corrosion of high-entropy alloys in NaCl solution. In 3.5% NaCl solution, the high-entropy alloy forms a passivation film during the polarization process, which is a very thin metal oxygen-containing compound [28,29]. The passivation film created on the surface of the high-entropy alloy has defects. For example, the native structure of the passivation film is loose, and Cl$^-$ can directly penetrate into the surface of the alloy to interact with the metal, thereby dissolving the alloy. At the same time, Cl$^-$ migrates along the grain boundaries of the passivation film; it locally dissolves and thins the surface passivation film during the migration process until it reaches the passivation film/metal interface and interacts with the metal, causing the passivation film to crack or even fall off, at which point the metal dissolves.

At the interface of the NaCl solution and the passivation film, the Cl$^-$ adsorbed on the surface of the passivation film forms a MeCl (or MeO(H)Cl) complex with a lower binding strength with the passivation film [30], and partial and continuous dissolution of the Cl-containing complexes. The adsorption amount of OH$^-$ at the dissolution of the passivation film is very small, which hinders the regeneration of the film and makes it continuously thin. When the passivation film is thinned, the migration of Cl$^-$ along the defect accelerates, and part of the Cl$^-$ preferentially reaches the surface of the alloy through the film defect and reacts with the metal to form chlorine-containing particles. The chlorine-containing particles continue to grow, and stress is generated between the passivation film and the metal. Since the bonding strength between the passivation film grains is smaller than that of the metal atoms, one side of the passivation film is broken, forming cracks, or even peeling off.

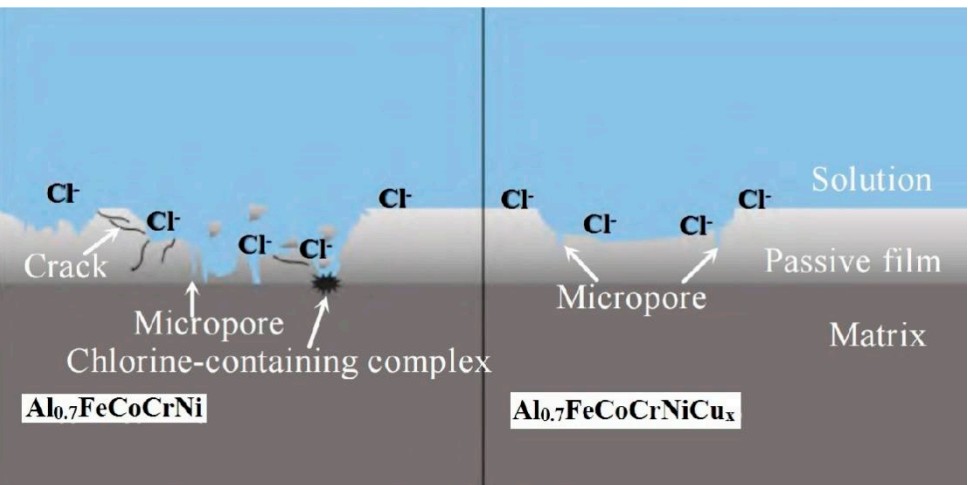

**Figure 5.** Schematic diagram of corrosion mechanism of the Al$_{0.7}$FeCoCrNiCu$_x$ high-entropy alloy.

In summary, the passivation film formed by the Al$_{0.7}$FeCoCrNi high-entropy alloy is not dense under the action of polarization, and a large number of continuous corrosion pits are formed. These pits are full of foam holes and even cracks (Figure 4a). The local thinning of the passivation film by Cl$^-$ and the stress caused by the Cl$^-$-containing particles at the interface of the passivation film/HEA lead to the rupture of the passivation film on the surface of the HEA. After the passivation film is ruptured, a large amount of Cl$^-$ forms direct contact with the metal and dissolves it, thereby greatly reducing the corrosion resistance. The Cu-containing Al$_{0.7}$FeCoCrNiCu$_x$ high-entropy alloy creates a denser passivation film with the isolated distribution of corrosion pits, and only foam-like holes are observed in the pits without crack generation (Figure 4e). Further, the local thinning of the passivation film by Cl$^-$ corrodes slowly.

### 4.4. Electrochemical Properties of Coatings

#### 4.4.1. Analysis of Dynamic Potential Polarization Curve

The corrosion potential (Ecorr) has no direct relationship with the corrosion rate of high-entropy alloys; it only characterizes the corrosion tendency of the alloy. The higher the corrosion potential, the lower the corrosion possibility. The corrosion current density (Jcorr) is a kinetic parameter that directly characterizes the corrosion rate of the alloy. The higher the corrosion current density, the higher the corrosion rate of the alloy and the more severe the corrosion [31]. The potential dynamic polarization curves of the Al$_{0.7}$FeCoCrNiCu$_x$ high-entropy alloys in 3.5% NaCl solution at room temperature are shown in Figure 6. Table 6 shows the corresponding electrochemical parameters.

**Table 6.** Electrochemical parameters.

| HEAs | Ecoor(vs.SCE)/mV | Jcorr/(A·cm$^{-2}$) |
|---|---|---|
| Al$_{0.7}$FeCoCrNi | −0.517 | 5.33 × 10$^{-6}$ |
| Al$_{0.7}$FeCoCrNiCu$_{0.30}$ | −0.486 | 4.40 × 10$^{-7}$ |
| Al$_{0.7}$FeCoCrNiCu$_{0.60}$ | −0.411 | 2.54 × 10$^{-7}$ |
| Al$_{0.7}$FeCoCrNiCu$_{0.80}$ | −0.401 | 2.03 × 10$^{-7}$ |
| Al$_{0.7}$FeCoCrNiCu$_{1.00}$ | −0.388 | 1.05 × 10$^{-7}$ |
| 5083 aluminum alloy | −0.522 | 5.73 × 10$^{-5}$ |

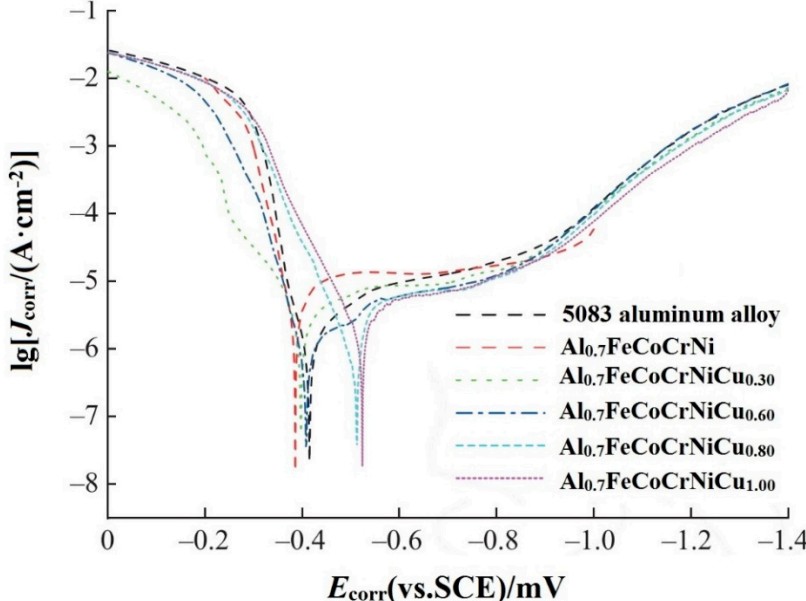

**Figure 6.** Potentiodynamic polarization curves of the Al$_{0.7}$FeCoCrNiCu$_x$ high-entropy alloys in 3.5%NaCl solution.

Figure 6 shows that $Al_{0.7}FeCoCrNi$ and $Al_{0.7}FeCoCrNiCu_{0.30}$ have obvious passivation behavior in the anode curve part. In the anodic curve, the active region is comparatively flat. As the potential increases, the curve enters the transition region, then passivates in the passivation region, and finally, turns to the over passivation region. In the polarization curves of $Al_{0.7}FeCoCrNiCu_{0.60}$, $Al_{0.7}FeCoCrNiCu_{0.80}$, and $Al_{0.7}FeCoCrNiCu_{1.00}$, no obvious passivation is found. This stems from the fact that the coating has a protective effect on surfaces with poor corrosion resistance, thereby reducing the corrosion rate.

It can be seen from Table 6 that when the molar ratio of Cu is 0.30, 0.60, 0.80, and 1.00, the corrosion potentials of the $Al_{0.7}FeCoCrNiCu_x$ high-entropy alloys are $-0.486$, $-0.411$, $-0.401$, and $-0.388$ V. This suggests that the corrosion potential Ecorr of the alloy gradually increases with the increase in Cu content. Among them, $Al_{0.7}FeCoCrNiCu_{1.00}$ has the highest self-corrosion potential, the lowest self-corrosion current density, the lowest corrosion rate, and the best corrosion resistance. This is because the $Al_{0.7}FeCoCrNiCu_x$ high-entropy alloy shows passivation behavior, and the Cu segregation is weak, which reduces the composition difference between the grain boundary and the grain. Moreover, it is not easy to form a galvanic cell, which hinders the corrosion of the $Al_{0.7}FeCoCrNiCu_x$ high-entropy alloy. The corrosion potential of the substrate is much lower than that of the coating, and the corrosion current density is one order of magnitude smaller than that of $Al_{0.7}FeCoCrNi$, indicating that the coating can effectively protect the substrate from corrosion. It is demonstrated that the high-entropy alloys containing Cu element have excellent corrosion resistance in NaCl solution, and Cu element can reduce the pitting susceptibility of the high-entropy alloy.

4.4.2. Analysis of Electrochemical AC Impedance Mapping

AC impedance mapping can be used to quantify the impedance magnitude of the passivation film, and thus, reflect the magnitude of the high-entropy alloy's corrosion resistance. Based on the electrochemical AC impedance results, the use of equivalent circuits to fit the alloy interfacial reaction model is an important tool to study the stability of passivation film. The electrochemical AC impedance spectrum, upon which the horizontal coordinate expresses the real part of the impedance $Z_{re}$ and the vertical coordinate, indicates the imaginary part of the impedance $Z_{im}$. The impedance profiles of the differently prepared alloys all show a capacitive arc compressed into a semicircular arc, which represents the bilayer capacitance of the solution interface and the electrode surface.

According to the test principle of impedance mapping, the size of the capacitive arc radius in the Nyquist diagram is related to the corrosion layer on the surface of the alloy and the material transfer of charge transfer. The size of the capacitive arc radius is approximately equal to the charge transfer resistance ($R_p$) during the corrosion reaction; moreover, the larger the capacitive arc radius, the greater the impedance and the better the corrosion resistance of the alloy. Figure 7a shows the AC impedance spectrum of the high-entropy alloy in 3.5 wt.% NaCl solution. Based on the graphical characteristics of the impedance spectrum (only one capacitive arc), an appropriate equivalent circuit diagram was chosen to fit the impedance spectrum, as shown in Figure 7b. In the figure, $R_s$ denotes the solution resistance between the reference electrode and the working electrode. $R_{ct}$ denotes the charge transfer resistance, characterizing the magnitude of the resistance of the anions and cations involved in the reaction to the discharge process at the electrode bilayer interface. Zw is the Warburg impedance, and Q is the constant phase angle element, indicating the double-layer capacitance of the working electrode and the corrosive medium, which is used to compensate for the effect caused by the inhomogeneity of the system instead of capacitance [32].

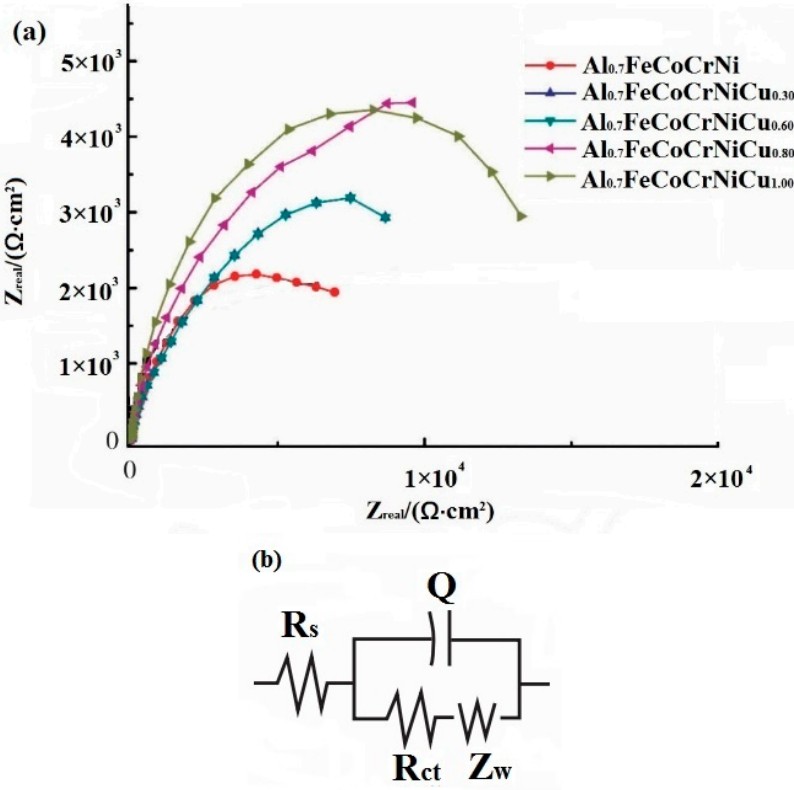

**Figure 7.** Nyquist plot (**a**) and equivalent circuit (**b**) of the $Al_{0.7}FeCoCrNiCu_x$ high-entropy alloy in 3.5 wt.% NaCl solution.

The electrochemical AC impedance test results were analyzed and fitted to the data using ZView2 software, and the results are presented in Table 7. As shown in Figure 7a, the radius of the alloy capacitive arc increases gradually with the increase in Cu element content, indicating that the impedance of the alloy surface becomes larger with the increment of Cu element. The increase in alloy surface resistance indicates the gradual increase in the corrosion resistance of the alloy. The charge transfer resistance can also further illustrate the corrosion resistance of the alloy. The higher the charge transfer resistance, the lower the current density and the lower the activity of the alloy surface, and, at the same time, the stronger the corrosion resistance of the alloy. As shown in Table 7, both charge transfer resistance (Rp) and impedance gradually increase with the increase in Cu elements, indicating that the corrosion resistance of the alloy is enhanced with the increase in Cu. The experimental results of AC impedance are consistent with the experimental results of the kinetic potential planning curve.

**Table 7.** AC impedance spectrum parameters of the $Al_{0.7}FeCoCrNiCu_x$ high-entropy alloy in 3.5 wt.% NaCl solution.

| Alloy | $R_s/(\Omega \cdot cm^2)$ | $Q/(mF/cm^2)$ | $R_p/(\Omega \cdot cm^2)$ |
|---|---|---|---|
| $Al_{0.7}FeCoCrNi$ | 1.15 | 0.12 | 5740 |
| $Al_{0.7}FeCoCrNiCu_{0.30}$ | 0.65 | 0.15 | 6190 |
| $Al_{0.7}FeCoCrNiCu_{0.60}$ | 0.78 | 0.12 | 7771 |
| $Al_{0.7}FeCoCrNiCu_{0.80}$ | 0.91 | 0.09 | 11,115 |
| $Al_{0.7}FeCoCrNiCu_{1.00}$ | 0.79 | 0.06 | 12,640 |

*4.5. Neutral Salt Spray Acceleration Test*

The neutral salt spray accelerated test is a corrosion resistance test employed for the artificially accelerated simulated corrosion of the $Al_{0.7}FeCoCrNiCu_{1.00}$ high-entropy alloy coating. In order to investigate the protective ability of the coating on the substrate after damage, the 5083 aluminum alloy substrate and the $Al_{0.7}FeCoCrNiCu_{1.00}$ high-entropy alloy coating samples were subjected to a neutral salt spray test at the same time. Before the experiment, the coating surface was scratched to damage the coating on the substrate. After the test, a layer of salt spray deposition was found on the surface of the test piece that was removed from the test chamber. Except for the deposited salt spray on the surface, the corrosion pattern on the surface of the test piece is completely consistent with the atmospheric corrosion pattern of aluminum alloys in the coastal humid and hot areas in the literature [33]. The corrosion products were rinsed, and the surface condition of the specimen was observed after drying. The results are shown in Figure 8.

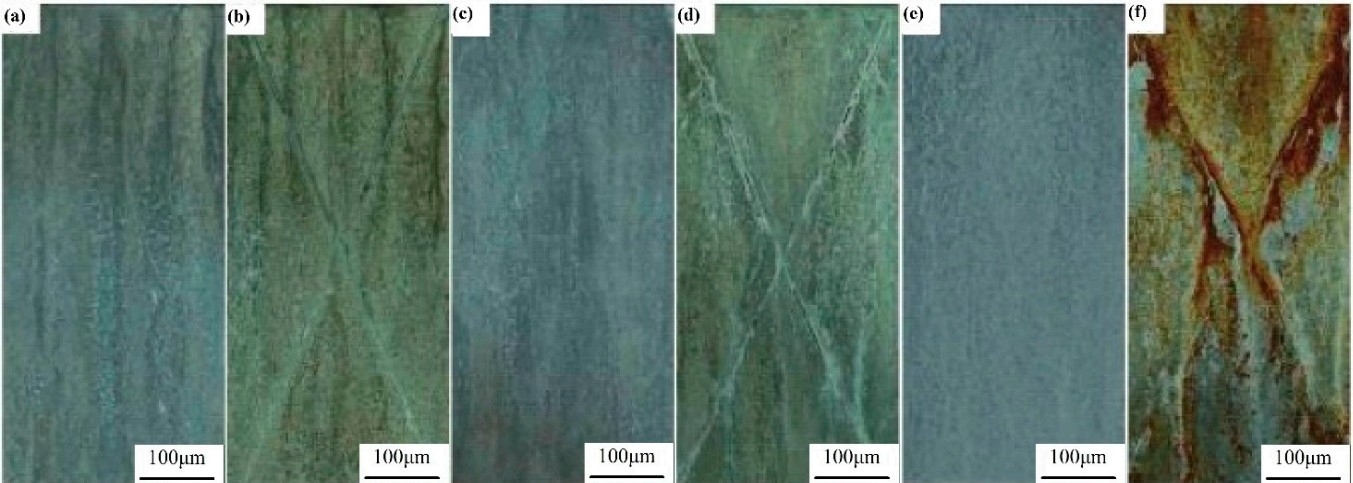

**Figure 8.** The neutral salt spray test of the $Al_{0.7}FeCoCrNiCu_{1.00}$ coating samples. (**a,b**) 1440 h; (**c,d**) 2880 h; (**e**) 4320 h; (**f**) 5040 h.

4.5.1. Qualitative Observation and Phenomenon Analysis of Accelerated Corrosion Test Pieces

Black rust was present on the entire 5083 aluminum alloy substrates after 2 h of the neutral salt spray acceleration test. After 1440 h, three corrosion traces appeared on the surface of the $Al_{0.7}FeCoCrNiCu_{1.00}$ high-entropy alloy coating, and there was a more uniform distribution of small white pitting pits; moreover, the increase in corrosion time saw an increase in the depth and diameter of the pitting pits. After 2880 h, the metal in the small area around the pitting pit of the coating remained bright, and the area around the larger pitting pit that was not corroded was also relatively large, indicating that the corrosion of aluminum alloy may be anodic dissolution. The area around the pit was protected by the cathode, and the inside of the pit contained anodic-accelerated corrosion. With an increase in corrosion time, the pitting pits continued to increase, and the surrounding bright area disappeared, replaced by a small pitting pit to stop the development. Black rust appeared on the surface of the scratch fork coating after 4320 h, which was examined as the corrosion product of the substrate, showing that the substrate can still be protected within 4320 h even if the coating is in a broken state. After 5040 h, some tiny pitting pits on the surface of the coating developed slowly or stopped; the corrosion process was concentrated on some large pitting pits, and the distribution density of the pitting pits was relatively small. That is, the amount of erosion pits decreased, but the radius and the depth of the pits increased significantly. This is because many of the original adjacent small pits developed and connected, and finally, converged into a large pitting pits that showed rapid corrosion. However, individual small pitting pits were located in the cathode of the electrochemical

process of the large pitting pits and are in a protected state, so the corrosion development is slow or stopped. From the specimen processing end face to observe corrosion, long-term corrosion specimen corrosion along the longitudinal and transverse development (flat grain boundary direction), corrosion traces narrow and long. The above observation results show that the accelerated corrosion of such a high-entropy alloy starts from pitting corrosion and develops into intergranular corrosion with the extension of corrosion time. The process of corrosion progress in this test and atmospheric corrosion [34], other accelerated corrosion tests and corrosion development processes, is basically the same, which can indicate the reliability of the acceleration method.

### 4.5.2. Quantitative Analysis of Accelerated Corrosion Test Pieces

(1)　Determination of corrosion weight loss

Figure 9 shows the variation in the corrosion weight loss of the $Al_{0.7}FeCoCrNiCu_{1.00}$ high-entropy alloy with time. Compared with the corrosion weight loss of 5083 aluminum alloy in the salt spray accelerated test [35], the solid line in the figure is the weight loss trend line. The accelerated environment used in [28] is the intermittent spraying of acid salt mist, and the continuous spraying method of acid salt mist is used in this paper. The figure shows that the average weight loss of the two acceleration methods is quite different in the short term after the initiation of corrosion. After a period of corrosion, the corrosion weight loss tends to be consistent in terms of size and development trend. The reason is, firstly, in the short term of the beginning of salt spray corrosion, due to the difference between the two acceleration methods of continuous spraying and intermittent spraying, the corrosion environment of the test piece is different, and the corrosion rate is different. Secondly, after a long period of spraying, a layer of salt spray deposits forms on the surface of the specimen. When the spraying stops intermittently, the temperature in the test box remains unchanged and the relative humidity is high. Due to the deposition of salt spray on the surface of the test piece, the small environment close to the surface of the test piece does not change much compared with the time of spraying. Furthermore, the two acceleration methods tend to be consistent.

The figure also shows that in the continuous spray accelerated corrosion test, the weight loss trend is close to a linear relationship, that is, the corrosion rate can reach a stable value faster than the intermittent spray, so the accelerated relationship of the accelerated corrosion test can be simpler.

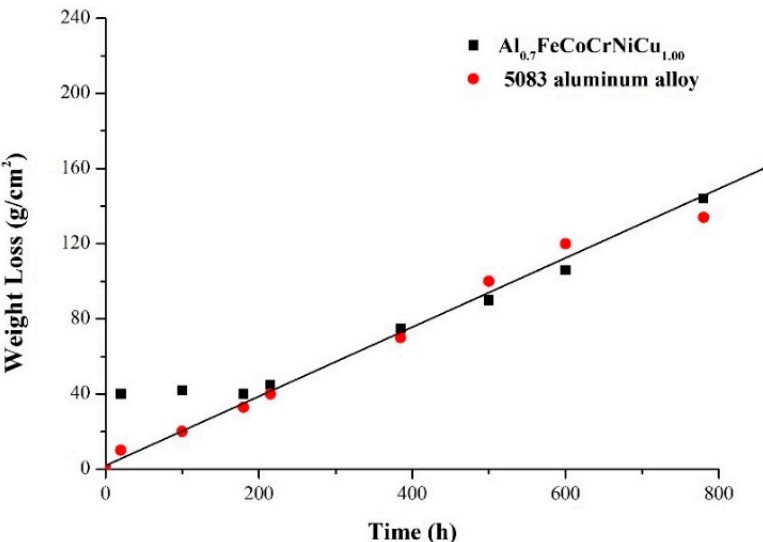

**Figure 9.** The weight loss varies with time.

(2) Determination of mechanical properties of corrosion specimens

Different from some previous mechanical property tests of pre-corroded specimens [36], in this paper, the dimensions of the test pieces were measured before salt spray corrosion and compared with the mechanical property parameters of the uncorroded test piece, in order to determine the effect of corrosion on the parameters. The decline of mechanical properties is shown in the Figure 10. It can be seen that the yield limit and the strength limit of the corrosion specimens decreased significantly. The straight line in the Figure 10 is the downward trend line of mechanical properties.

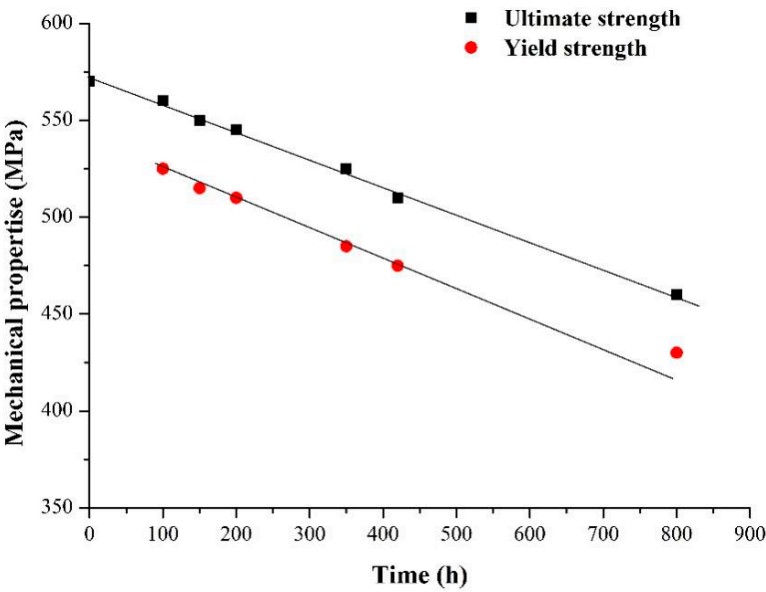

**Figure 10.** The decline of the samples' mechanical properties with corrosion time.

Figure 11 shows the compressive stress–strain curves at room temperature for specimens after corrosion of the high-entropy alloy $Al_{0.7}FeCoCrNiCu_x$ ($x$ = 0, 0.30, 0.60, 0.80, 1.00). The figure shows that when $x$ = 0, it has both elastic and plastic deformation, the compressive strength reaches 1315 MPa, the yield strength is 511 MPa, and the plastic deformation is 32.08%. When $x$ = 0.30, the compressive strength is 1619 MPa, which is significantly higher than that at $x$ = 0, but the plastic deformation is 5.99%, which is reduced relative to $x$ = 0. This is due to the large atomic radius of Cu, the addition of which leads to lattice distortion of the alloy, resulting in increased strength and decreased plasticity. With the increase in Cu content, the compressive strength and plastic deformation are the best at $x$ = 0.80, which are 1993 MPa and 11.43%, respectively. However, when $x$ = 1.00, the compressive strength and plastic deformation are reduced to 1772 MPa and 8.95%, respectively.

In summary, the neutral salt spray test result of the $Al_{0.7}FeCoCrNiCu_{1.00}$ high-entropy alloy coating is 5040 h, indicating that the corrosion resistance is higher than that of the substrate for more than 5000 h. In addition, the weight loss of the $Al_{0.7}FeCoCrNiCu_{1.00}$ high-entropy alloy coating increases in approximate proportion during the corrosion process, and the fracture strength and yield strength decrease in approximate proportion with the passage of time. With the increase in Cu content, the plasticity of the corroded $Al_{0.7}FeCoCrNiCu_x$ high-entropy alloy specimens first decreases and then increases, and the compressive strength first increases and then decreases. When $x$ = 0, its plasticity is best. When the content of Cu increases to $x$ = 0.80, its compressive strength and plasticity are the best.

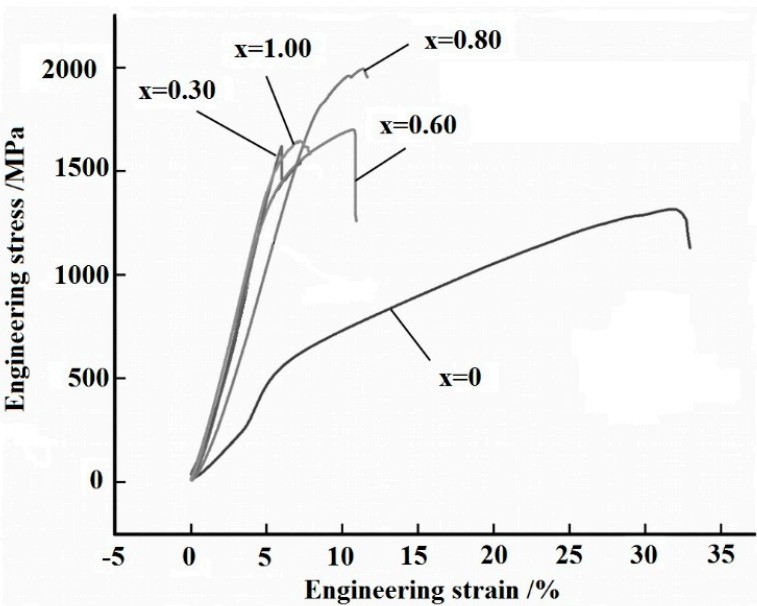

**Figure 11.** Compressive stress-strain curve at room temperature of the specimen after corrosion.

*4.6. Outdoor Atmospheric Exposure Test*

The outdoor exposure test is used to test the corrosion resistance of the $Al_{0.7}FeCoCrNiCu_{1.00}$ coating in the atmospheric environment. Exposure tests were carried out at the Beijing atmospheric test station and Qingdao Tuandao test station in an atmospheric environment, including the semi-rural atmosphere in the humid area of the southern temperate zone and the semi-industrial marine atmosphere in the humid area of the southern temperate zone. A total of 12 months of exposure experiments were carried out to compare and observe the surface corrosion of the $Al_{0.7}FeCoCrNiCu_{1.00}$ coating in different environments. The result is shown in Figure 12.

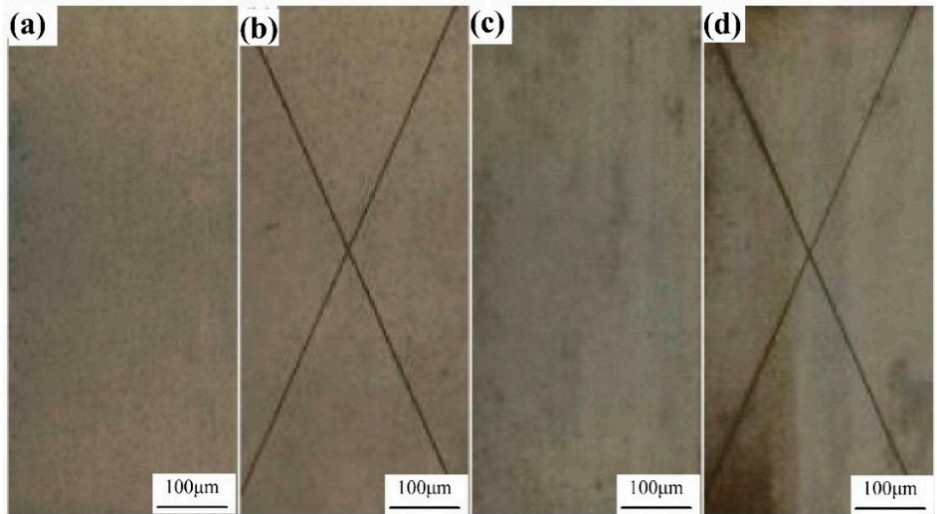

**Figure 12.** $Al_{0.7}FeCoCrNiCu_{1.00}$ coating specimens exposed to atmosphere for 12 months (**a,b**) Beijing; (**c,d**) Tuandao.

After 5083 aluminum alloy substrate was exposed in Beijing and Tuandao for 12 months, black rust appeared on the whole piece, while the surface of $Al_{0.7}FeCoCrNiCu_{1.00}$ coating was not obvious after 6 months of outdoor exposure in. After 12 months, the surface coating surface of the samples did not show any signs of corrosion.

According to the indoor neutral salt spray accelerated test and outdoor atmospheric exposure test results of 5083 aluminum alloy substrates and $Al_{0.7}FeCoCrNiCu_{1.00}$ coating, in the neutral salt spray accelerated test, the 5083 aluminum alloy substrate produced black rust within 2 h; the $Al_{0.7}FeCoCrNiCu_{1.00}$ coating protected the 5083 aluminum alloy substrate from corrosion for more than 5000 h; and even if the coating was damaged, the protection of the substrate could reach 4500 h. In the outdoor atmospheric exposure test, the 5083 aluminum alloy substrate was severely corroded after 12 months, and the $Al_{0.7}FeCoCrNiCu_{1.00}$ coating had no corrosion. This shows that the $Al_{0.7}FeCoCrNiCu_{1.00}$ coating has very good corrosion protection for 5083 aluminum alloy substrates and meets the application requirements in the marine atmospheric environment.

## 5. Conclusions

This paper details the microstructure of the $Al_{0.7}FeCoCrNiCu_x$ high-entropy alloy coating prepared on 5083 aluminum alloy substrates using the laser melting technique and the results of a corrosion resistance study in 3.5% (wt.%) NaCl solution. The observation of the $Al_{0.7}FeCoCrNiCu_x$ high-entropy alloy coating's microstructure revealed that with the increase in Cu content, the high-entropy alloy transformed from BCC1 and BCC2 to BCC1, BCC2, and FCC phases, which inhibited the formation of intermetallic compounds and the dissolution of passivation films caused by the dilution behavior of the matrix, resulting in the reduction in grain size and the increase in grain boundaries. The radius of capacitive arc resistance increased, the charge transfer resistance increased, the current density decreased, the surface activity of the alloy coating decreased, the corrosion rate decreased, and the corrosion resistance increased. In the continuous salt spray accelerated corrosion test with 3.5 wt.% NaCl solution (glacial acetic acid to adjust the pH value to 3.0) on the $Al_{0.7}FeCoCrNiCu_{1.00}$ high-entropy alloy, the corrosion resistance time reached 4320 h when the coating was damaged and over 5000 h when the layer was intact. Through the measurement of corrosion weight loss, the corrosion rate stability of this accelerated experiment proved to be excellent, which can provide reference significance for similar large-scale corrosion tests. The mechanical properties of the corroded specimens were tested, and it was found that the yield strength and strength limit were significantly decreased, and the corrosion weight loss, strength decrease, and damage development all showed linear changes. $al_{0.7}FeCoCrNiCu_{0.80}$ had the best compressive strength and plastic deformation. The outdoor atmospheric exposure test further proved that the $Al_{0.7}FeCoCrNiCu_{1.00}$ coating can guarantee the corrosion resistance of the substrate for more than 12 months, which can fully meet all the requirements for corrosion resistance of ships in the marine atmospheric environment.

The current research on the corrosion resistance of high entropy alloys mainly focuses on transition metal elements, and new forms of coating composition and properties can be explored by adding small amounts of rare-earth elements or rare-earth oxides to the coatings. Through optimization of the organization and composition, the design of more a comprehensive performance of high-entropy alloy coatings—especially the "high entropy effect" and "lattice distortion effect" on the corrosion performance of the coating research—is an important direction for future exploration of the corrosion mechanism.

**Author Contributions:** X.W. prepared the $Al_{0.7}FeCoCrNiCu_x$ high-entropy alloy coating. The micro morphology of the material before and after corrosion was photographed by a scanning electron microscope, and the XRD diffraction results, corrosion properties, and electrochemical properties of the coating were analyzed. At the same time, the neutral salt spray accelerated test was carried out, the test results were analyzed, and the manuscript was drafted and modified. Y.L. conducted outdoor atmospheric exposure experiments, compared and analyzed the experimental results of Beijing and Tuandao, and helped draft the manuscript. All authors have read and agreed to the published version of the manuscript.

**Funding:** This work was supported by the Youth Science and Technology Fund Projects of Gansu Province [grant number 20JR5RC608]; The Research Project on Teaching Reform of Vocational Education in Gansu Province [grant number 2021gszyjy-70]; and the "Fourteenth Five-Year" Educational Science Planning Project of Gansu Province [grant number GS [2021]GHB1772].

**Institutional Review Board Statement:** Not applicable.

**Informed Consent Statement:** Not applicable.

**Data Availability Statement:** The data used to support the findings of this study are included within the article.

**Conflicts of Interest:** The authors declare that there are no conflict of interest regarding the publication of this article.

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
