# Peer review of "Study on the Corrosion Resistance of Laser Clad Al0.7FeCoCrNiCux High-Entropy Alloy Coating in Marine Environment"

_coatings, doi:10.3390/coatings12121855_

Round 1
Reviewer 1 Report (New Reviewer)
The topic proposal in the manuscript sounds good, but the content of the article needs a complete rewrite.
There are a lot of spelling mistakes, scientific uncertainty, the proper description of the methods, and the wording is also doubtful in certain places.
Analytical measurements did not convince me either.
I have marked in yellow all the places that need to be changed. The abstract and conclusions should be reworded after correction.
I repeat myself, in its current state, the scientific status of the article does not even reach the minimum level, so a complete revision and rewriting is recommended.

Author Response
Please see the attachment.

Reviewer 2 Report (New Reviewer)
Wu et al. studied the corrosion resistance of ships based on laser cladding Al0.7FeCoCrNiCux high entropy alloy coating. The article is written nicely, explaining all the aspects related to the study. It will help researchers who are working in the same field. I recommend its acceptance after a major revision followed by the editorial correction.
1. Title should be modified as it does not give the scientific meaning of the research.
2. Abstract is nicely written with all the necessary information.
3. There should be a space (gap) between the numerical values and the units. Please check throughout the manuscript.
4. I have noticed that the authors used BOLD font in the middle of the paragraph, it seems the authors wish to highlight the key findings, but it is neither required nor recommended.
5. Figure 2 is not properly visible, please update it.
6. Figure 3(e) add 3 µm micrograph.
7. Figure 4 is also poor in quality, please update it.
8. The conclusion can be improved.
9. Please follow one homogeneous format style of referencing. Different styles of referencing can be seen.
10. Please check the entire manuscript and remove grammatical errors.
Author Response
Please see the attachment.

Reviewer 3 Report (New Reviewer)
The article presents and discusses in detail the results of the analysis of a coating made of a highly entropic Al0.7FeCoCrNiCux alloy prepared by by laser cladding technology on a substrate - an aluminum alloy 5083. The properties of the coating (coating microstructure, corrosion resistance, etc.) under different experimental conditions are presented here. I have no major comments on the presented results. I have only a formal remark, I believe that the citations are not given according to the guidelines for authors.
Author Response
Please see the attachment

Reviewer 4 Report (New Reviewer)
The manuscript titled "Study on Corrosion Resistance of Ships Based on Laser Cladding Al0.7FeCoCrNiCux High Entropy Alloy Coating" is a good attempt at work, however, needs a good improvement in the manuscript.
The authors need to address the below queries before considering this paper for possible publication:
1. The authors nowhere mentioned why the specific high entropy alloy is considered, which is very important with respect to the objectives of the manuscript. The authors need to discuss this in the introduction part without fail.
2. In the entire manuscript, the authors need to avoid the initials of the authors while citing in the running manuscript. Further, it is not necessary to explain where they have done this work (i.e. affiliations of the authors).
3. In PDP curves, why higher Cu-containing coatings have shown lower ECorr than in other cases? Need to explain.
4. why the cracks are observed in the coated samples where there is no Copper?
5. From Fig.9, the weight loss with time has no difference between coating and base 5083 alloy. Then what is the need for coating? Also, why the authors didn't provide data of weight loss for coating after 400h?
6. What is the need for the study of the mechanical properties with the varying times of exposure to a corrosive environment? If so, the authors need to give stress-strain diagrams also.
7. The references are not as per the guidelines. Moreover, there is no global coverage of reference citations. More than 80% are from a particular country. It would be better to have a global coverage.
8. Surprised to note both the BCC phases have same indexing points in their XRD pattern. Need a strong support for demonstrating the two separate BCC phases.
Round 2
Reviewer 1 Report (New Reviewer)
The quality of the manuscript is much better, thank you for the refining work.
Reviewer 2 Report (New Reviewer)
Accept
Reviewer 4 Report (New Reviewer)
Improved a lot.
This manuscript is a resubmission of an earlier submission. The following is a list of the peer review reports and author responses from that submission.
Round 1
Reviewer 1 Report
The authors have evaluated the corrosion resistance of ships based on laser cladding Al0.7FeCoCrNiCux high entropy alloy coating. This work is interesting and can be good for coating industries. Some serious issues exist in this work.
1. Several formatting issues and typing errors exist. For example, X-ray diffraction (XDR), paper Taking 7fecocrnicux, high bribe alloy, high smoke effect, etc. Please correct these.
2. Introduction section also needs to be re-written. There are many incomplete sentences and unclear statements. For example, “Postprocessing time, the corrosion resistance of epoxy composite sealing coating was better and more stable than that of sol-gel coating”.
3. What is the highlight of Figure 1? There is no scale bar.
4. The quality of Figure 2 is not good. Please add high resolution image here.
5. The sub-captions are not defined in Figure 5.
6. Same comment for Figure 6. Please improve.
7. Please check the unit of current density in Figure 7.
8. There are no scale bars in Figure 8 and 9. This is not scientific at all.
9. Section 5 should be discussed in detail.
Author Response
Response to Reviewer 1 Comments
I am very grateful to the reviewer for reviewing my manuscript during their busy schedule. I have made serious revisions in response to the revisions proposed by the teacher. Now the answers to the questions raised by the teacher are as follows:
Point 1:Several formatting issues and typing errors exist. For example, X-ray diffraction (XDR), paper Taking 7fecocrnicux, high bribe alloy, high smoke effect, etc. Please correct these.
Response1:I am very sorry for this error and apologize for the inconvenience caused to your reviewer.Part of the content has been deleted due to revision needs, and all possible spelling errors in the full text have been checked and revised.
Point 2: Introduction section also needs to be re-written. There are many incomplete sentences and unclear statements. For example, “Postprocessing time, the corrosion resistance of epoxy composite sealing coating was better and more stable than that of sol-gel coating”.
Response 2:The Introduction has been rewritten with the addition of recent developments in the study of the corrosion properties of HEA, and reworked for incomplete sentences and unclear formulations.
Point 3:What is the highlight of Figure 1? There is no scale bar.
Response3:The scale bar has been added to Figure 1 ,which is to illustrate the appearance of the 5083 aluminum alloy and Al0.7FeCoCrNiCu1.00 high-entropy alloy coatings before the experiment for comparison with subsequent test results.
Point 4:The quality of Figure 2 is not good. Please add high resolution image here.
Response 4:Figure 2 has been replaced with a higher resolution image.
Point 5:The sub-captions are not defined in Figure 5.
Response 5:Due to the logic of the expression, the order of some pictures in the manuscript is now adjusted, and the adjusted picture is shown in Figure 3. The sub-captions in Figure 3 have been re-labeled clearly.
Point 6:Same comment for Figure 6. Please improve.
Response 6:
Due to the need for logical expression, the original Figure 6 is now changed to Figure 4. The annotations in the figure have been modified. (a)-(e) are Al0.7FeCoCrNi, Al0.7FeCoCrNiCu0.30,Al0.7FeCoCrNiCu0.60,Al0.7FeCoCrNiCu0 .80, Al0.7FeCoCrNiCu1.00. On this basis, the title of Fig. 4 is further improved as SEM images of corrosion morphology of Al0.7FeCoCrNiCux high-entropy alloy coatings with different Cu contents to make the expression clearer.
Point 7:Please check the unit of current density in Figure 7.
Response 7:I am very sorry for this error, the current density units have now been revised.
Point 8:There are no scale bars in Figure 8 and 9. This is not scientific at all.
Response 8:I am very sorry for this kind of error. Due to the modification of the previous article, the original picture 8 is now picture 10, and the original picture 8 is now picture 11, and scale bars have been added.
Point 9:Section 5 should be discussed in detail.
Response 9:In order to express more clearly and accurately, Section 5 has been merged into Section 4, and the experimental phenomenon has been explained in detail and the reasons have been analyzed, and the corresponding conclusions have been drawn.

Reviewer 2 Report
The current manuscript does not contain novelty or impactful results. By reading the manuscript, we find it lacks many strengths that qualify it for publication in the journal. For these reasons, from my point of view, I reject publishing this article and attach some recommendations for authors.
Author Response
Thank you.
Reviewer 3 Report
Authors: Xuehong Wu and Helai Yang
Manuscript No: coatings-1905453
High entropy coatings with various Cu content are deposited on Al alloy substrate. Corrosion properties of the coatings are studied. The manuscript has several weak points:
1. The authors are encouraged to improve readability, style, and grammar of the manuscript. Some errors: "high bribe alloy", "Ming and Xiang", etc.
2. The introduction does not contain enough information on the past work of other researchers working on high entropy coatings.
3. Figure 2, I would like to ask the authors to index the individual peaks. What is the difference between BCC1 and BCC2? They have 3 common peaks in the figure. Do they have common lattice parameters?
4. Concerning Gibbs phase law, page 6, which parameters are known, and which are unknown? Why the authors use the law?
5. Captions of figs 3 and 4 are not clear.
6. Lines 193-199 need to improve clarity and strengthen evidence for making any conclusions.
Author Response
Response to Reviewer 3 Comments
I am very grateful to the reviewer for reviewing my manuscript during their busy schedule. I have made serious revisions in response to the revisions proposed by the teacher. Now the answers to the questions raised by the teacher are as follows:
Point 1: The authors are encouraged to improve readability, style, and grammar of the manuscript. Some errors: "high bribe alloy", "Ming and Xiang", etc.
Response 1: I am very sorry for this error, I have sought the help of a native English-speaking professional and made detailed corrections to the readability, style and grammar of the text, which greatly improved the readability of the text.
Point 1:The introduction does not contain enough information on the past work of other researchers working on high entropy coatings.
Response 2:The latest progress in the research of corrosion properties of high entropy alloys has been added in lines 55-98 of the introduction, which constitutes a complete introduction to the research at home and abroad.
Point 3: Figure 2, I would like to ask the authors to index the individual peaks. What is the difference between BCC1 and BCC2? They have 3 common peaks in the figure. Do they have common lattice parameters?
Response 3: The indexs of the individual peaks have been added to Figure 2.The BCC1 structure is composed of CrNi, AlNi, CrNi2, Co3Ni2, etc., the BCC2 structure is composed of Fe3Cr2, Fe2Cr and other structures, and the FCC phase is mainly FeNi3 and FeNi2.The five curves in the figure correspond to the same Al0.7FeCoCrNiCux high-entropy alloy material, so the positions of the BCC1 and BCC2 phases do not change without considering the influence of the Cu element on the structure in each curve.With the increase of Cu element, the BCC1 and BCC2 phases in the high-entropy alloy gradually decrease, and the FCC phase gradually increases, but it does not affect the change of their positions.So there are 3 common peaks.
Point 4:Concerning Gibbs phase law, page 6, which parameters are known, and which are unknown? Why the authors use the law?
Response 4:The manuscript mentions the Gibbs phase law P=C+1-F (P is the number of phases, C is the principal component, and F is the thermodynamic degree of freedom), C=5, F=2. This law is mentioned only to show that the general alloy phase number, principal component number and thermodynamic degrees of freedom satisfy this law, but high-entropy alloys do not satisfy this law due to the particularity of their structure, and explain the reasons for not satisfying them.Therefore, it is necessary to make the microstructure of Al0.7FeCoCrNiCux high-entropy alloy.
Point 5:Captions of figs 3 and 4 are not clear.
Response 5:The title of Figure 3 "Typical microstructure of high entropy alloy coating" has been revised to "Microstructure of Al0.7FeCoCrNiCux-based high-entropy alloy coatings with different Cu contents";The title of Figure 4 "Corrosion morphology of high entropy alloy coating" was modified to "SEM images of corrosion morphology of Al0.7FeCoCrNiCux high-entropy alloy coatings with different Cu contents".
Point 6: Lines 193-199 need to improve clarity and strengthen evidence for making any conclusions.
Response 6:The added content has been reflected in lines 278-340 of the text. The supplementary content elaborates the corrosion mechanism of Al0.7FeCoCrNiCux high entropy alloy in detail, providing a clear and reliable basis for reaching the final conclusion.
Round 2
Reviewer 1 Report
The authors have done a great job. The comments have been addressed.
Author Response
I would like to thank the reviewers for their recognition of my revision work, thank you very much.
Reviewer 3 Report
The authors improved their manuscript significantly. There are still some minor issues:
Line 45 accelerat -->accelerate
Lines 48,49: In Mg-Mn-Ce an alloy or a ceramic layer?
Line 73: .and --> and
Lines 191,193: what is the meaning (35+1)?
Lines 238,239: Sentence "With the addition of..." contradicts with Fig. 2 and Lines 268,269
Line 270: I recommend that the authors leave out word "very".
Lines 287, 288: Is "point-like" and "punctate" the same type of structure? If yes, please unify the usage.
Line 589: I assume that boundaries between grains and grain boundaries are the same. If so, please use only one of them.
